# A Recombinant Genotype I Japanese Encephalitis Virus Expressing a *Gaussia* Luciferase Gene for Antiviral Drug Screening Assay and Neutralizing Antibodies Detection

**DOI:** 10.3390/ijms232415548

**Published:** 2022-12-08

**Authors:** Chenxi Li, Xuan Chen, Jingbo Hu, Daoyuan Jiang, Demin Cai, Yanhua Li

**Affiliations:** 1College of Veterinary Medicine, Yangzhou University, Yangzhou 225009, China; 2Comparative Medicine Research Institute, Yangzhou University, Yangzhou 225009, China; 3Jiangsu Co-Innovation Center for Prevention and Control of Important Animal Infectious Diseases and Zoonoses, Yangzhou 225009, China; 4College of Animal Science and Technology, Yangzhou University, Yangzhou 225009, China

**Keywords:** genotype I Japanese encephalitis virus, *gaussia* luciferase, antiviral drug screening, neutralization assay, reporter virus

## Abstract

Japanese encephalitis virus (JEV) is the major cause of viral encephalitis in humans throughout Asia. In the past twenty years, the emergence of the genotype I (GI) JEV as the dominant genotype in Asian countries has raised a significant threat to public health security. However, no clinically approved drug is available for the specific treatment of JEV infection, and the commercial vaccines derived from the genotype III JEV strains merely provided partial protection against the GI JEV. Thus, an easy-to-perform platform in high-throughput is urgently needed for the antiviral drug screening and assessment of neutralizing antibodies specific against the GI JEV. In this study, we established a reverse genetics system for the GI JEV strain (YZ-1) using a homologous recombination strategy. Using this reverse genetic system, a *gaussia* luciferase (Gluc) expression cassette was inserted into the JEV genome to generate a reporter virus (rGI-Gluc). The reporter virus exhibited similar growth kinetics to the parental virus and remained genetically stable for at least ten passages in vitro. Of note, the bioluminescence signal strength of Gluc in the culture supernatants was well correlated with the viral progenies determined by viral titration. Taking advantage of this reporter virus, we established Gluc readout-based assays for antiviral drug screening and neutralizing antibody detection against the GI JEV. These Gluc readout-based assays exhibited comparable performance to the assays using an actual virus and are less time consuming and are applicable for a high-throughput format. Taken together, we generated a GI JEV reporter virus expressing a Gluc gene that could be a valuable tool for an antiviral drug screening assay and neutralization assay.

## 1. Introduction

Japanese encephalitis virus (JEV) is a mosquito-borne flavivirus that belongs to the *Flavivirus* genus of the *Flaviviridae* family, and it is a highly pathogenic zoonotic virus that causes serious neurologic disease in humans [1]. According to the World Health Organization (WHO), an estimated three billion people from over 24 countries are currently living in the JEV epidemic areas [2,3,4], and there are as many as 68,000 JEV cases per year with a mortality rate of 10–15% [5,6]. In the zoonotic transmission cycle of the JEV, rice paddy-breeding *Culex tritaeniorhynchus* act as the primary vector, wading birds and pigs serve as the primary amplifying/reservoir host, and humans are considered to be the dead-end host of JEV infection [7,8]. JEV could invade the human central nervous system and eventually results in permanent neurological damage and sequelae, or even death [9,10]. Until now, no clinically approved drug is available for the specific treatment of JEV infection. Although vaccination is the most effective strategy to prevent JEV infection, the incidence of JEV is still increasing.

Like other flaviviruses, the JEV has a single-stranded positive-sense RNA genome of approximately 11 kb in length. The JEV genome comprises a single open reading frame (ORF) flanked by 5′ and 3′ untranslated regions (UTR) and encodes a polyprotein precursor processed subsequently by viral and host proteinases to generate three structural proteins (envelope [E], pre-membrane [prM], and capsid [C]) and seven nonstructural proteins (NS1, NS2A, NS2B, NS3, NS4A, NS4B, and NS5) [11]. Phylogenetic investigations of the nucleotide sequence of the envelope (E) gene indicated that the JEV has five geographically and epidemiologically distinct genotypes (genotype I to V), and most isolates were classified as genotype I (GI) or genotype III (GIII) [12,13]. The GIII JEV strain was first isolated in 1935 and has been the predominant genotype associated with outbreaks in most Asian countries [14]. The number of the GI JEV isolated in the field has increased dramatically in the past twenty years [15]. Of note, the GI almost completely replaced the GIII as the dominant genotype in Asia, especially in China [16,17].

The emergence of the GI strains as the dominant genotype has raised concerns about the urgency for specific antiviral drugs to treat GI JEV infections and the re-evaluation of the protective efficacy of licensed Japanese encephalitis vaccines derived from the GIII strains [18,19]. Thus, there is an urgent need for an easy-to-perform platform in high-throughput for the antiviral drug screening and assessment of neutralizing antibodies. Benefiting from the development of the reverse genetics system, reporter viruses have been generated and widely utilized for tracking viral infections and quantifying viral replication in vitro and in vivo [20,21]. For flavivirus, the generation of a recombinant virus expressing a reporter gene, such as fluorescent proteins (eGFP, mCherry) and luciferase proteins (*firefly* and *Renilla* luciferases; Fluc and Rluc) [19,22,23,24], is a common strategy for virus tracking and also provides a rapid and convenient tool for the screening of antiviral drugs and the evaluation of neutralizing antibodies. However, recombinant viruses carrying large reporter genes are usually genetically unstable [25,26]. Considering the flavivirus capsid (C) coding region could tolerate the insertion of foreign genes, the reporter protein was usually inserted between two copies of the capsid gene. To address the instability issues, the coding region of the duplicated capsid gene was codon optimized to stabilize the viral genome [25,27].

*Gaussia* luciferase (Gluc) from the marine copepod *Gaussia princeps* is a small molecule with a higher bioluminescent signal than the widely used Fluc and Rluc. Since Gluc is naturally secreted from mammalian cells into the culture medium, its expression level can be quantified by a luciferase assay using a culture supernatant without lysing cells [28]. In this study, we constructed an infectious cDNA clone of the GI JEV strain YZ-1 using a homologous recombination strategy. Using this reverse genetic system, a Gluc expression cassette was inserted into the JEV genome to generate a reporter virus expressing Gluc, rGI-Gluc. This reporter virus exhibited similar growth kinetics to the parental virus and remained genetically stable for at least ten passages in vitro. The bioluminescence signal strength of Gluc secreted into the culture media of the reporter virus-infected cells were well correlated with the virus titers. With this reporter virus, we established novel Gluc readout-based assays for the antiviral drug screening and neutralizing antibody detection against the GI JEV.

## 2. Results

### 2.1. Construction and Recovery of a Gluc-Tagged GI JEV

The full-length cDNA clone of the GI JEV YZ-1 strain was constructed as depicted in Figure 1A. Four fragments (F1, F2, F3, and F4) covering the complete JEV genome were amplified by RT-PCR and one step assembled into the linearized pOK12-HDVR vector to obtain the cDNA clone pOK-rGI. With this cDNA clone, the transcription of the JEV RNA genome is under the control of a T7 promoter and the hepatitis D virus ribozyme (HDVR) following the JEV genome ensures the generation of an authentic 3′ terminus (Figure 1A). The genomic sequences encoding the C_1_ to C_34_ of the capsid protein have been confirmed to be dispensable for viral replication [20,27]. To generate a GI JEV reporter virus stably expressing Gluc, we chose the site between the C_34_ and C_35_ residues in the capsid for the insertion of a Gluc expression cassette which includes the coding sequences of a Gluc gene, a T2A peptide, and a duplicated N-terminal 34 amino acid (C*_1–34_) that has been codon-optimized as previously reported (Figure 1B) [19].

The full-length genomic RNAs of the parental clone and the reporter clone generated by in vitro transcription were transfected into BHK-21 cells to rescue viruses, respectively. At 36 hpt, the typical cytopathic effect (CPE) characterized by cell shrinkage and death was observed in cells with RNA transfection (Figure 1C). Furthermore, viral replication in BHK-21 cells inoculated with the rescued parental virus (rGI) or the reporter virus (rGI-Gluc) was confirmed by an IFA against the JEV NS3 protein (Figure 1D). Meanwhile, we confirmed that Gluc expressed by the rGI-Gluc can be secreted into the culture supernatant (Figure 1E). An extremely high level of luciferase activity up to 1.8 × 10^7^ luminance units was detected in the 20 μL culture supernatant of the rGI-Gluc-infected cells but not in the culture supernatants of the mock-infected and rGI-infected cells. In addition, we confirmed the correct insertion of the Gluc expression cassette in the rGI-Gluc by verification of the modified genomic regions through RT-PCR amplification and DNA sequencing.

### 2.2. In Vitro Characterization and Genetic Stability of a Gluc-Tagged GI JEV

To determine whether the insertion of Gluc affects the replication ability of the GI JEV, we compared the growth kinetics of the rGI-Gluc with those of the rGI through multiple-step growth curves. Both the rGI-Gluc and rGI reached their peak titers around 10^6.1^ TCID_50_/mL at 60 hpi (Figure 2A). Their similar growth kinetics were also confirmed by their similar expression levels of the viral NS3 protein at 24, 36, 48, and 60 hpi and similar plaque morphology and size (Figure 2B,C). The bioluminescence signal strength of Gluc in the culture supernatant of the reporter virus-infected cells was well correlated with the virus titers by TCID_50_ (Figure 2A). Therefore, the Gluc activity in the culture supernatant could serve as a good indicator of the reporter virus infection.

Next, we serially passaged the rGI-Gluc ten times in BHK-21 cells. The genetic stability of the rGI-Gluc was analyzed by the verification of the integrity of the Gluc expression cassette in the P1, P5, and P10 viruses. The viral RNAs were extracted and subject to RT-PCR amplification using specific primers (Table 1) anchoring the 5′ UTR and capsid protein coding region. Consistent with the expected sizes of the amplicons, a single DNA fragment with approximately 1.1 kb in length was amplified with the cDNA of P1, P5, or P10 as the template (Figure 2D), suggesting that the Gluc expression cassette could remain stable for at least 10 passages in BHK-21 cells. Furthermore, the growth kinetics of the P1, P5, and P10 viruses were evaluated with TCID_50_ assays and a *gaussia* luciferase assay, respectively. The P1, P5, and P10 viruses exhibited a very similar growth trend based on the virus titer and luciferase activity in the culture supernatants (Figure 2E).

### 2.3. Applicability of Gluc-Tagged GI JEV for Antiviral Drug Screening

As there are no clinically approved drugs for the specific treatment of JEV infection, a convenient and accurate high-throughput screening system (HTS) is urgently required. In this study, we assessed the application of the rGI-Gluc in the antiviral assay against the GI JEV. BHK-21 cells were inoculated with the rGI-Gluc or rGI at an MOI of 0.1 and treated with various concentrations of nitroxoline, a potent inhibitor of the JEV [19,26]. At 36 hpi, the inhibitory effect of the nitroxoline against the rG-1 and rGI-Gluc was measured by an IFA, TCID_50_ assay, and *gaussia* luciferase assay. As expected, nitroxoline significantly inhibited the rGI and rGI-Gluc growth in a dose-dependent manner (Figure 3A), and no detectable cytotoxicity caused by nitroxoline was observed at the corresponding concentrations (Figure 3B). Based on the virus titration results, the IC_50_ of nitroxoline against the rGI-Gluc was 3.293 μM which was similar to that of the rGI (IC_50_ = 4.266 μM) (Figure 3B). We also examined the inhibitory effect of nitroxoline on the growth of the rGI-Gluc based on the Gluc activities. The luciferase activity in the culture supernatant of the rGI-Gluc-infected cells was gradually reduced with the increasing concentration of nitroxoline (Figure 3C), and the IC_50_ was calculated to be 5.681 μM (Figure 3D). Thus, the sensitivity of the rGI-Gluc to nitroxoline is similar to that of the parental virus, and the reduction in the Gluc activity in the culture supernatant can be used interchangeably with the reduction in the virus titer in the antiviral drug screening assay against the JEV.

We validated the application of the rGI-Gluc in an antiviral drug screening assay using three compounds, ARDP0006, 27-hydroxycholesterol, and ribavirin, which exhibit antiviral activity against other viruses [29,30,31]. ARDP0006 and ribavirin possessed a potent inhibitory effect on virus propagation based on the reduction in the luciferase activity, whereas no obvious inhibition effect of 27-hydroxycholesterol on the JEV was observed (Figure 3E). The IC_50_ values of ARDP0006 and ribavirin against the JEV were 1.302 μM and 37.56 μM, and no detectable cytotoxicity was detected at the corresponding concentrations (Figure 3E).

### 2.4. Detection of NAb against GI JEV Based on a Novel Luciferase-Based Serum Neutralization Assay

As all current commercial vaccines were derived from the GIII JEV strains and merely provided partial protection against the other genotypes of the JEV [18,32], it is necessary to re-evaluate the levels of neutralizing antibodies against the GI JEV induced by the GIII JEV vaccine. Traditionally, the determination of the JEV NAb titers has mainly relied on an IFA and a plaque reduction neutralization test (PRNT), which are cumbersome and time consuming. In this study, we developed a novel luciferase readout-based serum neutralization assay as illustrated in Figure 4A. Using this neutralization assay, we evaluated the titers of cross-neutralizing antibodies against the GI JEV reporter virus. The NAb titer of a serum sample was expressed as the reciprocal of the highest dilution of a sample causing a 90% reduction in the Gluc activity. The NAb titers against the rGI-Gluc in the four sera were 2^3^~2^5^ and lower than 2 in the rest sera (Figure 4B). The NAb titers were also confirmed by the IFA detection of the NS3 protein (Figure 4C). Furthermore, we evaluated the NAb titers against the GIII JEV (HSY-1 strain) in these samples using the fluorescence readout-based neutralization assay. As expected, a higher percentage (80%) of sera exhibited a NAb titer above 2, and in the four sera with neutralizing antibodies against both genotypes, higher NAb titers were detected against the GIII JEV (Figure 4C).

## 3. Discussion

Japanese encephalitis is one of the most common forms of viral encephalitis in humans around the world, especially in Asia and Western Pacific regions [33,34]. In the last two decades, the emergence of the genotype I JEV as the dominant genotype in Asian regions has raised a significant threat to public health [35,36]. Until now, no clinically approved drug is available for the specific treatment of JEV infection, and all current commercial vaccines derived from the GIII JEV strains merely provided partial protection against the GI JEV [18,32]. Thus, it is urgent to establish an accurate, reproducible, and high-throughput method for antiviral drug screening. Benefiting from the development of reverse genetics systems, the development of reporter flaviviruses has greatly advanced in recent years. Multiple types of reporter genes, such as fluorescent (EGFP, mCherry) and luciferase (Rluc, Fluc) tags, have been widely used to track viral infections and quantify viral replication in vitro and in vivo [21,24,27]. In this study, we constructed a GI JEV reverse genetic system via homologous recombination in vitro and generated a recombinant reporter virus by engineering a Gluc expression cassette into the GI JEV backbone. Using this reporter virus, we developed a novel high-throughput approach based on a luciferase readout for an antiviral drug screening and serum neutralization assay.

The selection of a suitable site for the insertion of a reporter gene is critical for the construction of the reporter flaviviruses. There are three options in the flavivirus genome for the insertion of a gene of interest, including the site between the E and NS1 coding regions [37], the site between the 5′ UTR and capsid gene [20], and the site between the NS5 gene and the 3′ UTR [38]. However, the insertion of foreign genes tends to destabilize the flavivirus genome, which often results in the loss of reporter genes during serial passage in vitro [26,39]. To date, the site between the 5′ UTR and capsid gene exhibited the best tolerance for the insertion of a foreign gene [20] and was also opted as the site for the reporter gene insertion in this study. Considering that it contains the indispensable *cis*-acting RNA elements required for genomic cyclization [40,41], the JEV genomic sequence encoding N-terminal 34 residues of the capsid protein were added to the N-terminus of Gluc. To avoid the homologous recombination between the C_1–34_ and C*_1–34_, the duplicated genomic sequences (C*_1–34_) coding N-terminal 34 amino acids were codon-optimized. The 2A sequence from thosea asigna virus (T2A) is a “self-cleaving” small peptide that enables the separation of two proteins encoded by a single mRNA through the ribosomal skipping mechanism at this peptide sequence. Thus, the T2A peptide following the reporter virus enables the release of Gluc from the capsid protein. Eventually, the reporter virus expressing Gluc (rGI-Gluc) was successfully rescued and remained genetically stable for at least ten passages in vitro.

The insertion of the reporter genes, such as Rluc and Fluc genes, in the flavivirus genomes often impairs the growth ability of the resultant reporter viruses [24]. Gluc is naturally secreted from mammalian cells into the culture medium and can generate over 1000-fold higher bioluminescent signals than Rluc and Fluc. In comparison with other reporter genes, the Gluc gene is smaller in size, which will reduce the size of the foreign gene insertion and may make the resultant reporter virus more stable. As a reporter gene, Gluc has been widely applied for the development of reporter viruses [42,43,44]. As expected, the GI JEV reporter virus expressing Gluc generated in this study exhibited similar growth properties to its parental virus, and the expression levels of Gluc in the virus supernatants were correlated well with the viral titers. Since the rGI-Gluc remained genetically stable and effectively secreted Gluc into the culture supernatant during infection, we evaluated this reporter virus for an antiviral assay against the JEV using nitroxoline, which is a known JEV inhibitor [19]. The infection of the rGI-Gluc was suppressed by the nitroxoline treatment in a similar dose-dependent manner based on the reduction of the viral titers and Gluc levels in the culture supernatants. Additionally, the sensitivity of this reporter virus to nitroxoline is similar to its parental virus. We also tested this luciferase readout-based antiviral drug screening assay with ARDP0006, 27-hydroxycholesterol, and ribavirin.

Vaccination remains the most effective strategy for preventing JEV infection [45]. Vaccine-induced neutralizing antibodies play a key role in protection against JEV infection, and the titers of neutralizing antibodies are essential for this protection [46,47]. Since all JEV live attenuated vaccines and inactivated vaccines are derived from the genotype III strains, it is necessary to re-evaluate the levels of neutralizing antibodies against the genotype I JEV induced by vaccination. Currently, the PRNT and IFA are standard methods for the detection of JEV-specific Nabs. However, they are time consuming, labor-intensive, and not applicable for high-throughput formats. Here, a luciferase readout-based neutralization assay for the GI JEV was developed in a high-throughput format. The SA14-14-2 vaccine is widely used for JEV control in both humans and pigs [45]. We utilized this luciferase readout-based neutralization assay to evaluate the NAbs titers against the GI JEV in several serum samples collected from mice immunized with the SA14-14-2 vaccine. The results were very similar to those obtained with the current gold standard immunofluorescence assay, suggesting that this luciferase readout-based assay is accurate and reliable in evaluating neutralizing antibodies against the GI JEV. Notably, in comparison with the NAb levels against the GIII JEV, the NAb titers against the GI JEV were much lower in mice vaccinated with a commercial vaccine, which is consistent with the previous reports that the GIII JEV-derived vaccine could provide partial protection against the infection of the GI JEV [18,32,48].

To conclude, we successfully constructed a GI JEV reporter virus stably expressing a secretory Gluc. Furthermore, this reporter virus is a valuable tool for a high-throughput drug screening and neutralization assay against the GI JEV.

## 4. Materials and Methods

### 4.1. Cells, Viruses, Antibodies, and Vector

Baby hamster kidney (BHK-21) cell line purchased from American Type Culture Collection (ATCC; Manassas, VA, USA) was cultured at 37 °C in Dulbecco’s modified Eagle’s medium (DMEM; Hyclone, Logan, UT, USA) supplemented with 10% fetal bovine serum (FBS; Sigma, St Louis, MO, USA), penicillin (100 U/mL), and streptomycin (100 μg/mL). The GI JEV strain of YZ-1 (GenBank accession number: MZ540901) and GIII JEV strain of HSY-1 (GenBank accession number: MZ540902) were isolated and stored in our laboratory, and all viruses were propagated and titrated in BHK-21 cells. A polyclonal rabbit antibody against JEV NS3 protein was purchased from Genetex (St. Anthony, TX, USA). Alexa 488-conjugated goat anti-rabbit IgG H&L was purchased from Jackson ImmunoResearch Inc. (West Grove, PA, USA). The low-copy plasmid pOK-12 was purchased from BioVector NTCC Inc. (Beijing, China).

### 4.2. Construction of a Full-Length cDNA Clone of GI JEV YZ-1 Strain by Homologous Recombination In Vitro

Initially, a synthesized DNA fragment containing a hepatitis D virus ribozyme (HDVR) sequence was chemically synthesized in vitro (GenScript, Nanjing, China) and cloned into a pOK-12 plasmid to create the pOK12-HDVR vector. To construct the full-length cDNA clone of the GI JEV YZ-1 strain, the complete JEV genome was divided into four overlapping fragments which were inserted into the pOK12-HDVR vector through homologous recombination technology, as illustrated in Figure 1A. In brief, the total RNA of BHK-21 cells infected with YZ-1 was extracted with the FastPure Viral DNA/RNA Mini Kit (Vazyme Biotech, Nanjing, China) and reverse transcribed into cDNA using the Superscript IV first-strand synthesis system (ThermoFisher Scientific, Waltham, MA, USA). According to the full-length genomic sequence of the JEV YZ-1 strain, four pairs of primers (Table 1) were designed to amplify four overlapping DNA fragments (F1, F2, F3, and F4) covering the complete viral genome with the viral cDNA as a template, in which the F1 fragment contained a T7 promoter sequence and adjacent individual DNA fragments shared homology arm sequences of approximately 20 nucleotides. Ultimately, the F1, F2, F3, and F4 fragments were synchronously assembled into the linearized pOK12-HDVR vector linearized with restriction enzyme *Not I,* using the NEBuilder HiFi DNA Assembly Master Mix (NEB, Ipswich, MA, USA). The full-length cDNA clone of the JEV YZ-1 strain was designated as pOK-rGI.

### 4.3. Construction of a cDNA Clone Containing a Luciferase Gene

As illustrated in Figure 1B, a Gluc gene was inserted into the cDNA clone pOK-rGI by homologous recombination. Briefly, the DNA fragment S1 (contains the T7 promoter, viral 5′ UTR, and the first 102 nt of C gene) and S2 (contains 103 to 1200 nt of viral polyprotein ORF) were amplified using the specific primers listed in Table 1. The expression cassette of Gluc (Gluc-T2A-C_34_) consisting of Gluc coding sequence, a 2A peptide of thosea asigna virus (T2A), and the codon-optimized N-terminal 102 bp of C gene was synthesized in vitro (GenScript, Nanjing, China). As depicted in Figure 1B, the fragments of S1, S2, and Gluc-T2A-C_34_ were assembled into the linearized pOK-rGI using the NEBuilder HiFi DNA Assembly Master Mix (NEB, Ipswich, MA, USA), and the resultant plasmid was designated as pOK-rGI-Gluc.

### 4.4. Recovery of the Recombinant Viruses

To rescue the recombinant viruses, the pOK-rGI and pOK-rGI-Gluc plasmids linearized by *SalI* digestion were used as templates for T7-directed in vitro transcription using the mMESSAGE mMACHINE T7 Transcription Kit (Invitrogen, Carlsbad, CA, USA), respectively. After treatment with *DNaseI* to remove the DNA template, the RNA transcripts were transfected into BHK-21 cells on a six-well plate by using the DMRIE-C reagent (ThermoFisher Scientific, Waltham, MA, USA). At 36 to 60 h post-transfection (hpt), the culture supernatants were collected, clarified by centrifugation, and passaged to fresh BHK-21 cells. The recombinant viruses were plaque-purified three times in BHK cells and subsequently confirmed by Sanger sequencing.

### 4.5. Viral Growth Kinetics and Plaque Assay

BHK-21 cells at 80% confluency in a 24-well plate were infected with the parental virus and the recombinant viruses at a multiplicity of infection (MOI) of 0.01, respectively. After 1 h of incubation at 37 °C, unadsorbed viruses were washed off with PBS and fresh DMEM medium with 2% FBS was added. At 24, 36, 48, 60, and 72 h post-infection (hpi), culture supernatants were harvested and titrated by the TCID_50_ assays on BHK-21 cells. The multiple-step growth curves were created with GraphPad Prism 8.0 software according to the Reed–Muench method. For plaque assay, BHK-21 cells in 6-well plates with 90% confluence were infected with JEV at the dose of 100 to 200 TCID_50_ per well and incubated for 2 h at 37 °C. The cells were washed three times with PBS and cultured in modified Eagle’s medium (MEM; Gobico) supplemented with 2% FBS and 1% low melting point agarose for 4 days. The cells were fixed with 4% paraformaldehyde for 30 min at room temperature and then stained with 0.5% crystal violet to visualize plaques.

### 4.6. Indirect Immunofluorescence Assay and Western Blot Analysis

BHK-21 cell monolayers infected with the parental viruses or the recombinant viruses were fixed with 4% paraformaldehyde for 30 min at room temperature. For NS3 protein detection, the cells were first blocked with 1% BSA and then incubated with a polyclonal rabbit antibody against JEV NS3 protein (1:400 dilution with PBS) at 37 °C for 1 h. After three washing with PBS, the cells were stained with Alexa 488-conjugated goat anti-rabbit IgG H&L, and cell nuclei were counterstained with 4′,6-diamidino-2-phenylindole (DAPI) solution (Solarbio Life Sciences, Beijing, China). For western blot analysis, cell lysates collected with the RIPA lysis buffer (Beyotime Biotechnology, Shanghai, China) were separated in SDS-PAGE gels and then transferred onto a nitrocellulose (NC) membrane. NC membranes were blocked with 5% skim milk in PBS for 2 h at room temperature and further incubated with the following primary antibodies: anti-JEV NS3 protein (1:5000 dilution with PBS); anti-Actin (1:5000 dilution with PBS; Proteintech, Chicago, IL, USA) overnight at 4 °C. Subsequently, the membranes were washed three times with PBST (PBS containing 0.05% Tween-20) and incubated with HRP-conjugated secondary antibodies for 1 h at room temperature. Protein bands were detected using an ECL reagent (Vazyme Biotech, Nanjing, China) according to the manufacturer’s instructions.

### 4.7. Gaussia Luciferase Assay

The *gaussia* luciferase assay was performed as previously described [42]. Briefly, culture supernatants were harvested and cleared by centrifugation at 4 °C. The luciferase activity was determined in a luminometer (LumiStation-1800; Flash, Shanghai, China) by mixing 20 μL supernatant of cell samples with 50 μL reaction substrate Coelenterazine h (20 μM, pH 7.2) which was diluted into PBS supplemented with 5mM NaCl.

### 4.8. Cell Viability Assay and Antiviral Assay

The cytotoxicity of Nitroxoline, 1,8-Dihydroxy-4,5-dinitroanthraquinone (ARDP0006), 27-hydroxycholesterol, and ribavirin (Millipore, Billerica, MA, USA) was assessed by cell counting kit-8 (CCK-8) assay according to the manufacturer’s instructions (Vazyme Biotech, Nanjing, China). BHK-21 cell Monolayers in a 96-well plate were treated with each drug at indicated concentrations, and dimethyl sulfoxide (DMSO) treatment was used as the negative control. At 48 h later, 100 μL 10% CCK-8 reagent per well was added and incubated for 1 h at 37 °C. The absorbance was measured at 450 nm using a microplate reader (Gallop Technology, Shanghai, China) and the 50% cytotoxic concentration (CC_50_) was calculated by using the software GraphPad Prism 8.0.

The application of rGI-Gluc in the antiviral drug screening assay was subsequently investigated. BHK-21 cell monolayers in 96-well plates at 80% confluency were inoculated with rGI-Gluc at an MOI of 0.1 in a culture medium supplemented with serially diluted Nitroxoline, ARDP0006, 27-hydroxycholesterol, or ribavirin. After 1 hour of incubation, the inoculum was removed and the freshly prepared medium supplemented with drugs was added at the indicated concentrations. At 36 hpi, culture supernatants were collected and subject to luciferase assay. The 50% inhibition concentrations (IC_50_) were calculated using GraphPad Prism 8.0 software.

### 4.9. Serum Neutralization Assay

Ten serum samples collected from 4-week-old C57BL/6 mice that were immunized with genotype III JEV vaccine (SA14-14-2) were stored in our laboratory. The serum samples were inactivated at 56 °C for 30 min and were serially 2-fold diluted with DMEM. Subsequently, 50 μL serum samples at different dilutions were mixed with an equal volume of rGI-Gluc, rGI, or HSY-1 strain at the dose of 200 TCID_50_/50 μL. After 1 h’s incubation at 37 °C, the mixture of virus and serum (100 μL/sample) was added to a cell monolayer in a 96-well culture plate. After incubation for 1 h at 37 °C, cell monolayers were washed three times with PBS and maintained in the infection medium of DMEM supplemented with 2% FBS. At 24 hpi, JEV replication was evaluated by an IFA or luciferase assay. The titers of neutralizing antibody (NAb) against genotype I JEV were calculated using the Reed–Muench method.

### 4.10. Biosafety Procedures and Practices

The Institutional Biosafety Committee (IBC) of Yangzhou University has approved all biological experiments for JEV in this study. All experiments in this study involving authentic viruses were performed in the biosafety level 2 facility of Yangzhou University, strictly following all biosafety management and regulations.

## Figures and Tables

**Figure 1 ijms-23-15548-f001:**
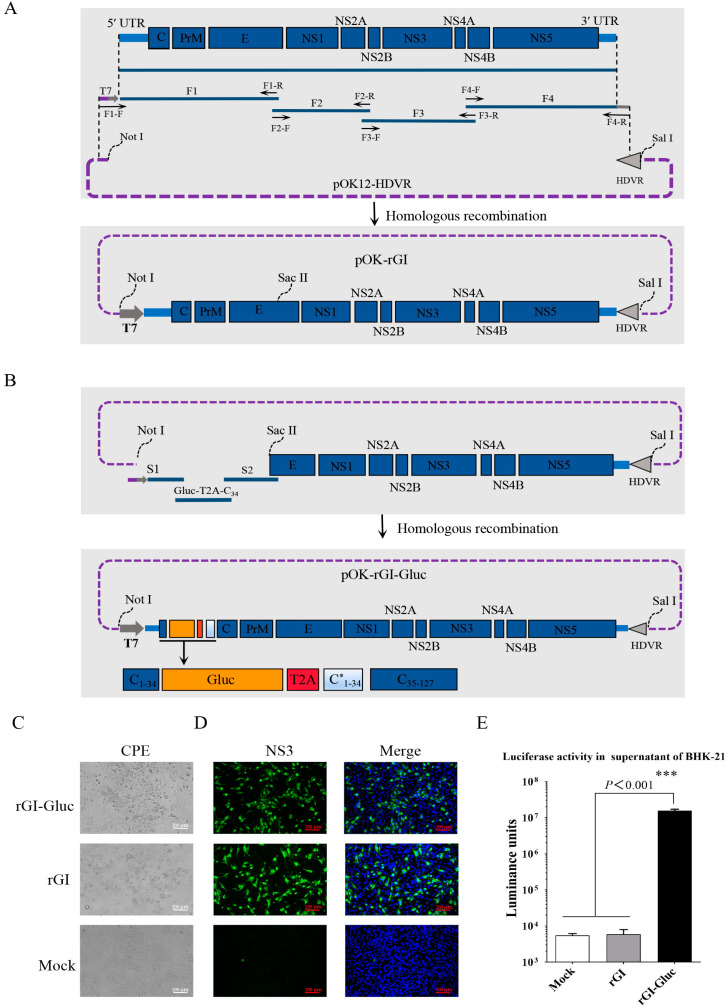
Construction and recovery of Gluc-tagged GI JEV. (**A**,**B**) Strategy for the construction of the infectious cDNA clone of GI JEV strain YZ-1 and Gluc-tagged GI JEV. T7, T7 promoter. HDVR, hepatitis D virus ribozyme sequence. (**C**) The cytopathic effects in BHK-21 cells transfected with viral RNA at 36 hpi. (**D**) Verification of the rescued JEV by IFA. BHK-21 cells were respectively infected with a parental virus (rGI) and a reporter virus (rGI-Gluc) for 24 h. The expression of JEV NS3 protein was detected with an antibody against NS3 (green), and cell nuclei were stained with DAPI (blue). (**E**) Luciferase activity in culture supernatants. The supernatants of BHK-21 cells mock-infected or infected with viruses were respectively harvested and analyzed by luciferase assay. All data are presented as mean ± SD from three independent experiments and tested by Student’s *t*-test. ***, *p* < 0.001.

**Figure 2 ijms-23-15548-f002:**
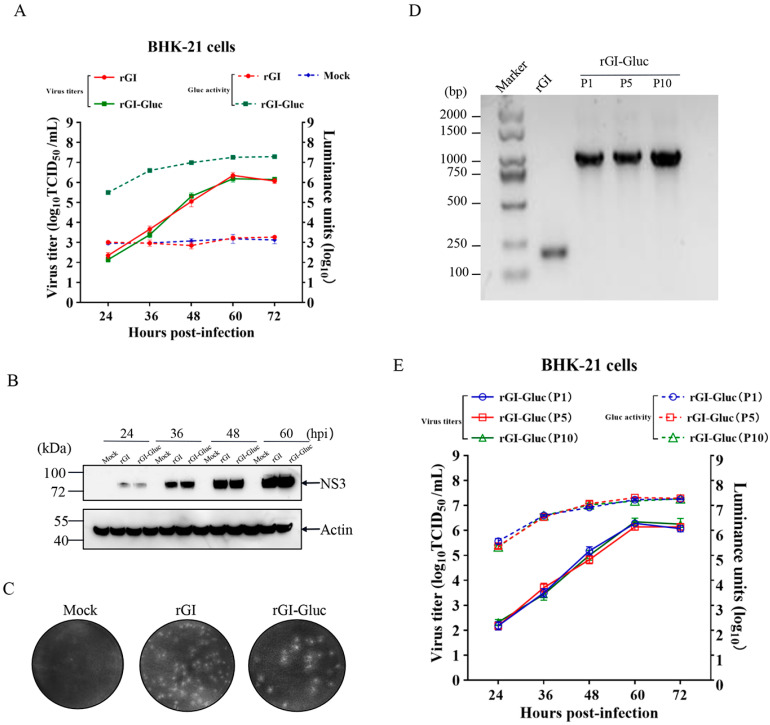
Characterization and genetic stability of Gluc-tagged GI JEV. (**A**) The multi-step growth curves of the parental virus and the reporter virus. The BHK-21 cell monolayers were respectively infected with rGI-Gluc or rGI at an MOI of 0.01. The supernatants were harvested to titrate by TCID50 assays in BHK-21 cells or subject to luciferase assay at indicated time points after infection. All data are presented as mean ± SD from three independent experiments. (**B**) The expression of NS3 protein in infected BHK-21 was determined by western blot analysis. The cell lysates at the indicated time points were harvested and subject to western analysis with an anti-NS3 antibody. β-actin was detected as a loading control. NS3 and β-actin were indicated by black arrows. (**C**) The plaque morphology formed by rGI-Gluc and rGI in BHK-21 cells. The BHK-21 cells were respectively infected with rGI or rGI-Gluc at a dose of 200 TCID50. At 4 dpi, the cells were fixed with 4% paraformaldehyde and the plaques were visualized by crystal violet staining. (**D**) RT-PCR amplification of viral genomic region covering the Gluc expression cassette in rGI-Gluc. The corresponding genomic region in rGI was also amplified using the same pair of primers. The expected sizes of amplicons are 1.1 kb and 0.22 kb for rGI-Gluc and rGI. (**E**) The multi-step virus growth curves of rGI-Gluc of P1, P5, and P10 are based on viral progenies and Gluc activities in culture supernatants.

**Figure 3 ijms-23-15548-f003:**
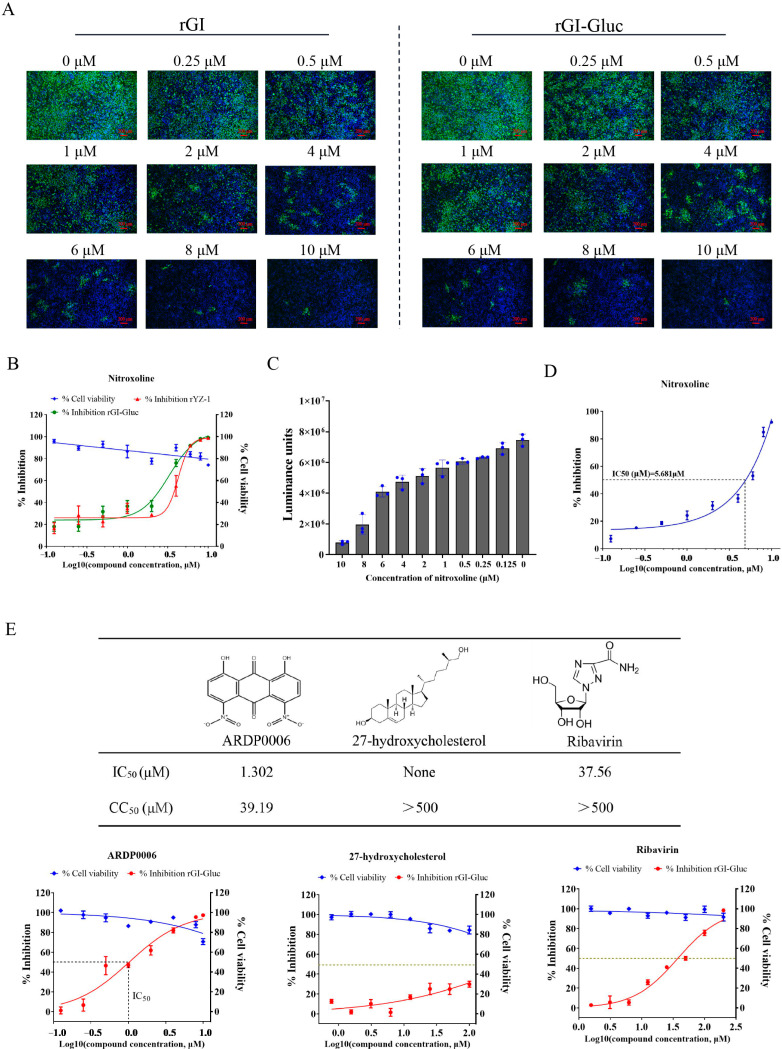
Application of Gluc-tagged GI JEV in the antiviral assay. (**A**) Antiviral effect of nitroxoline on rGI and rGI-Gluc infection. BHK-21 cells were respectively infected with rGI and rGI-Gluc at an MOI of 0.1 and simultaneously treated with serial two-fold dilutions of nitroxoline. At 36 hpi, the expression of NS3 protein in virus-infected cells was detected by IFA, and cell nuclei were stained with DAPI. (**B**) The antiviral effect of nitroxoline on the infection of rGI-Gluc and rGI in BHK-21 cells based on viral titer reduction. The cytotoxicity of nitroxoline on BHK cells at the indicated concentrations was measured by CCK-8 assay. The values represent the mean ± SD from three experiments. (**C**,**D**) The antiviral effect of nitroxoline on the infection of rGI-Gluc was determined by the reduction in the Gluc activities. The 50% inhibitory concentration (IC50) values were calculated by the Log (dose)-response curves. (**E**) Antiviral profiles of three compounds (ARDP0006, 27-hydroxycholesterol, and ribavirin) against rGI-Gluc in BHK-21 cells.

**Figure 4 ijms-23-15548-f004:**
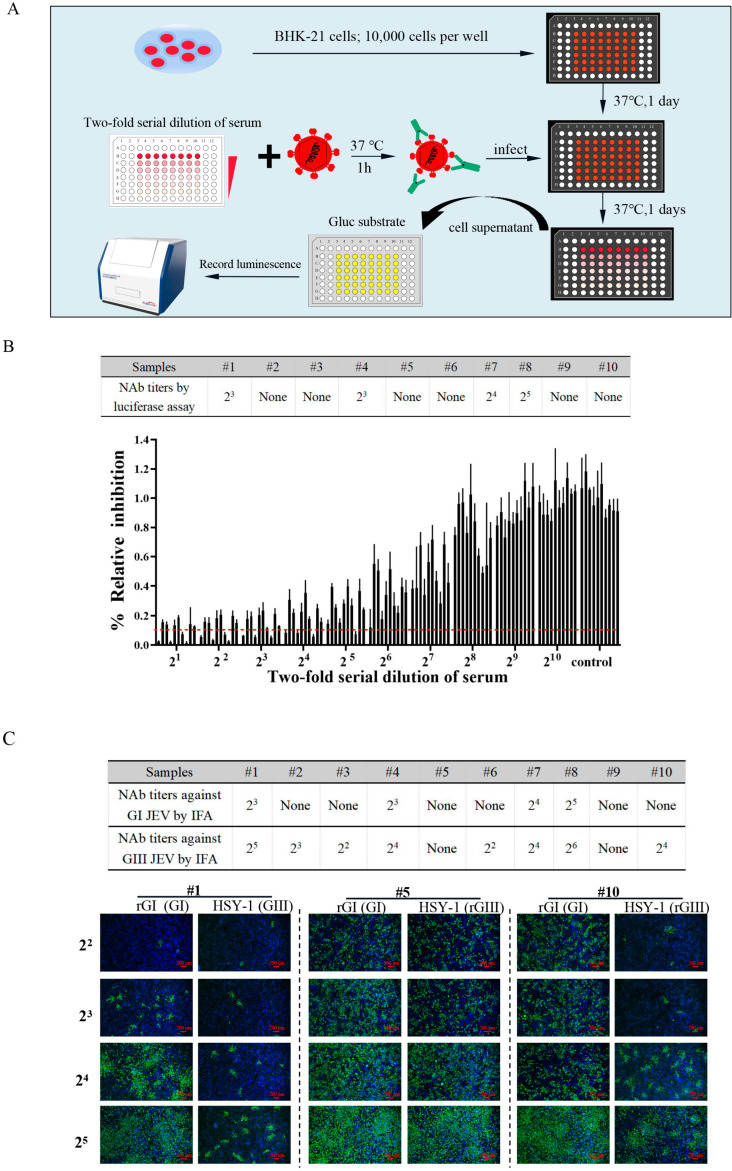
Application of Gluc-tagged GI JEV in the neutralization assay. (**A**) A schematic diagram of the procedure for serum neutralization assay based on Gluc readout. (**B**) The titers of neutralizing antibodies in mice serum samples. Inactivated two-fold serially diluted serum was mixed with an equal volume of rGI-Gluc at the dose of 200 TCID50. After 1 h incubation at 37 °C, the mixture was added to BHK-21 cell monolayers in a 96-well culture plate and incubated for an additional 1 h at 37 °C. At 24 hpi, the cell supernatants were harvested and subject to luciferase assay. Each data point represents the mean ± SD from three experiments. (**C**) The NAb titers against rGI (GI) and HSY-1 (GIII) in mice sera.

**Table 1 ijms-23-15548-t001:** The primers used in this study.

Primers Name	Sequence	Usage
F1-F	tatggaaaaacggctttggcggccgc**taatacgactcactatagg**agaagtttatctgtgtgaacttcttggc	Amplification of the F1 or S1 fragment
F1-R	ttgtgtgatccaagacattcccccaaagag	
F2-F	ctctttgggggaatgtcttggatcacacaa	Amplification of the F2 fragment
F2-R	tggaacaccgggatcatcaatcaagtgaaa	
F3-F	tttcacttgattgatgatcccggtgttcca	Amplification of the F3 fragment
F3-R	ggcttgtcagcgttcttgatgagagtcca	
F4-F	tggactctcatcaagaacgctgacaagcc	Amplification of the F4 fragment
F4-R	gaggtggagatgccatgccgacccagatcctgtgttcttcctcaccac	
C34-Gluc-R	tttgactcccatcccggaaactaccctcttcactccca	Amplification of the S1 fragment
Gluc-T2A-F	gaatcccggcccttccgggatgaccaagaagccaggaggcccg	Amplification of the S2 fragment
Gluc-T2A-R	Ttcccgaaaagtccacatccatttcc	
Gluc-F	aggagggcccggaaaaaaccgggccat	Detection of Gluc gene
Gluc-R	Ataatcaagctttcattccctcctc	

Underlined sequences are the T7 RNA polymerase promoter sequences.

## Data Availability

Not applicable.

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
