# Peer review of "A Recombinant Genotype I Japanese Encephalitis Virus Expressing a Gaussia Luciferase Gene for Antiviral Drug Screening Assay and Neutralizing Antibodies Detection"

_ijms, 2022, doi:10.3390/ijms232415548_

Round 1

Reviewer 1 Report

In this study, the authors have constructed a gaussia luciferase (Gluc) expressing reporter virus for genotype-I Japanese encephalitis virus (g-I JEV) using a reverse genetic system. Finally, using this reporter virus, the authors have performed antiviral drug screening and tested the neutralizing antibodies of g-III JEV against Gluc g-I JEV.

The primary concern of this study is, using the same approach, the reporter viruses have been constructed already for the Japanese encephalitis virus, including for genotype-1 (PMID: 34973281, 32750466), except the authors have inserted a gaussia luciferase (Gluc) reporter gene.

The manuscript is well-written, and the experimental design is good. I have a few comments.

1.     Line number 23- at least passages- ten passages?

2.     Fig 2A – Gluc activity – wrong labeling

3.     T2A peptide – the authors can add a few more lines to describe the role of this self-cleaving peptide

4.     Please make sure the references are cited correctly. For example, (line 39) there are as many as 68,000 JEV cases per year, with a mortality rate of 10%~15%. 68,000 cases per year was reported by Campbell et al. (PMID: 22084515)

Author Response

1. Line number 23- at least passages- ten passages?

Authors’ Response:  Thank you for the correction. We have modified 'at least passages’ to ‘at least ten passages’.

2. Fig 2A – Gluc activity – wrong labeling

Authors’ Response: Thank you for the correction. We have corrected this error in Fig 2A in the revision.

3. T2A peptide – the authors can add a few more lines to describe the role of this self-cleaving peptide.

Authors’ Response:  Thank you for the suggestion. We have added a few more lines to describe the role of T2A self-cleaving peptide in discussion .

4. Please make sure the references are cited correctly. For example, (line 39) there are as many as 68,000 JEV cases per year, with a mortality rate of 10%~15%. 68,000 cases per year was reported by Campbell et al. (PMID: 22084515)

Authors’ Response:  Thank you for the suggestion. We have carefully proofread the references in the manuscript to ensure the correct citation.

Reviewer 2 Report

This is very interesting data and would be useful in the field.

Author Response

Thanks for your comments on our manuscript.

Reviewer 3 Report

I personally find this work interesting, well designed, and well documented. The manuscript is clear, and presented in a well-structured manner.

A crucial point to accept for pubblication would be to add the experimental evidence that the Gaussia luciferase (Gluc) expression cassette has not integrated into the genome of BHK-21 cells transfected with viral RNA  at 60 hpi or more. 

This experimental evidence can be obtained by PCR from or same template of Fig 2 panel D without retrotrascription step or by extracting the DNA from cells infected and maintained in culture for 60 hpi or more generations. The negative results could esclude any integration in genomic DNA BHK-21 cells of a viral genome manipulated with insertion of a humanized foreign gene (Gluc expression cassette).

Moreover, I consider also an important point in order to accept the manuscript for publication that the Authors consider to add at least a paragraph that is currently missing entitled  “biosafety procedures and practices” (in material and methods section of the submitted manuscript) in which they  provide clear information and specification of procedures implemented during handling and for disposal of the biological material and viruses employed considering the possibility or impossibility of accidental leaks. 

Please, check title editing of the first reference (at Line 440 and 441) and also the omogeneity in the text of citations, and moreover,  I suggests also to add following references

-van den Hurk, A.F.; Skinner, E.; Ritchie, S.A.; Mackenzie, J.S. The Emergence of Japanese Encephalitis Virus in Australia in 2022: Existing Knowledge of Mosquito Vectors. Viruses 2022, 14, 1208. https://doi.org/10.3390/v14061208
-Solomon T, Dung NM, Kneen R, et al Japanese encephalitis Journal of Neurology, Neurosurgery & Psychiatry 2000;68:405-415.
-Solomon, Tom, Haolin Ni, David W. C. Beasley, Miquel Ekkelenkamp, Mary Jane Cardosa, and Alan D. T. Barrett. “Origin and Evolution of Japanese Encephalitis Virus in Southeast Asia.” Journal of Virology 77, no. 5 (2003): 3091–98. doi:10.1128/JVI.77.5.3091-3098.2003.

Author Response

1.  A crucialpoint to accept for pubblication would be to add the experimental evidence that the Gaussia luciferase (Gluc) expression cassette has not integrated into the genome of BHK-21 cells transfected with viral RNA at 60 hpi or more. 

Authors’ Response:  We are very grateful for your suggestions. As well known, the JEV genome consists of a single-strand positive sense RNA, and the entire replication cycle of the virus completely occur in the cytoplasm, but not in cell nucleus (Nat Rev Microbiol, 2005. DOI: 10.1038/nrmicro1067). In addition, Japanese encephalitis virus (JEV) belongs to the flavivirus, but not to retrovirus, and all flaviviruses could not produce proviral DNA integrated into the host genome during viral replication cycle. Therefore, we believe that the Gaussia luciferase (Gluc) expression cassette could not be integrated into the genome of BHK-21 cells.

2. I consider also an important point in order to accept the manuscript for publication that the Authors consider to add at least a paragraph that is currently missing entitled  “biosafety procedures and practices” (in material and methods section of the submitted manuscript) in which they provide clear information and specification of procedures implemented during handling and for disposal of the biological material and viruses employed considering the possibility or impossibility of accidental leaks. 

Authors’ Response:  Thank you for the suggestion. We have added a paragraph about ‘biosafety procedures and practices’ in material and methods .

3. Please, check title editing of the first reference (at Line 440 and 441) and also the omogeneity in the text of citations, and moreover, I suggest also to add following references.

Authors’ Response: Thank you for the suggestion. We have carefully proofread the references in the manuscript to ensure the correct citation. We have added the references suggested by this reviewer in our manuscript.